# Simulation of Onset of the Capillary Surface Wave in the Ultrasonic Atomizer

**DOI:** 10.3390/mi12101146

**Published:** 2021-09-23

**Authors:** Yu-Lin Song, Chih-Hsiao Cheng, Manoj Kumar Reddy, Md Saikhul Islam

**Affiliations:** 1Department of Bioinformatics and Medical Engineering, Asia University, Taichung 413, Taiwan; 2Department of Computer Science and Information Engineering, Asia University, Taichung 413, Taiwan; manoj14a0kumar@gmail.com (M.K.R.); mdsaikhulislam@gmail.com (M.S.I.); 3Zeng Hsing Industrial Co., Taichung 411, Taiwan; forcea9931@gmail.com

**Keywords:** capillary wave, surface wave, harmonic, subharmonic, COMSOL, finite element analysis, ultrasonic atomizer

## Abstract

The novel drug delivery system refers to the formulations and technologies for transporting a pharmaceutical compound in the body as it is needed to safely achieve its desired therapeutic effects. In this study, the onset vibrational amplitude of capillary surface waves for ultrasonic atomization spray is explained based on Faraday instability. Using ultrasonic frequency, the vibrational amplitude approached a critical point, and the liquid surface broke up into tiny drops. The micro-droplets were are steadily and continuously formed after the liquid feeding rate was optimized. The simulation study reported a minimum vibrational amplitude or onset value of 0.38 μm at 500 kHz frequency. The required minimum energy to atomize the drops was simulated by COMSOL Multiphysics simulation software. The simulation result agreed well with the numerical results of a subharmonic vibrational model that ocurred at 250 kHz frequency on the liquid surface. This newly designed single frequency ultrasonic atomizer showed its true physical characteristic of resonance on the fluid surface plane. Hence, this research will contribute to the future development of a single-frequency ultrasonic nebulizer and mechatronics for the generation of uniform atomized droplets.

## 1. Introduction

Ultrasonic atomization is a useful technology and has many practical applications for industrial processes such as combustion of fuel, spray drying, spray cooling, nano-emulsion film coating, and the fabrication of printed electronics and sensors, as well as for miniature carriers in pharmaceuticals [1,2]. The capillary wave became of great interest in the middle of the 20th century, and the atomization of incompressible frictionless fluid initiated based on this principle is known as atomized liquid droplets [3]. The capillary surface wave studies are derived from Faraday’s instability, which was proposed in 1831 [4]. The atomization of fluid is caused by capillary waves formed in an unstable state, and small droplets are formed by a collapsing of unstable surface waves. Control of the droplet process and characterization of the size of the vaporized droplets are both crucial to obtaining a product of the desired quality [5]. Biomedical engineering has implemented this technique for drug delivery and gene therapy, and there is still much to be elucidated [6,7]. It has also been applied to material drying, due to the easy process and high yield. Ultrasonic atomized nozzles use high-frequency vibrations produced by piezoelectric crystal driving transducers, which act upon the nozzle tip to create capillary waves in a liquid film. At threshold amplitude, the horizontal flat-equilibrium fluid surface turns into a capillary surface wave, and then starts breaking off into tiny liquid droplets [8]. The factors that influence the initial droplet size are the surface tension, frequency of vibration, and viscosity of the liquid. According to Perron’s study, the predictable and uniform water droplets are formed under a sufficient oscillatory vibration. Thus, the larger vibrational amplitude causes cavitation due to the agglomeration of the droplets after leaving the unstable horizontal surface waves [9,10]. A vibrator container holding a liquid is used for ultrasonic atomization. The vibrator is placed at the bottom of the container, and the liquid inside the container experiences a vertical vibration that propagates waves on the stationary surface of the water. This destabilizes the horizontal surface of the water accumulated by the nonlinear standing wave groups [11].

Benjamin and Ursell used Mathieu’s equation to create a stability diagram for a preferable incompressible frictionless liquid. If the characteristic wave number and amplitude of the vibration (p, q) are noticed in the stability chart, this indicates that liquid fluid is unstable. Therefore, this implies that the surface is stable without regular periodic waves [12]. The numerical analysis and atomization experimental progress confirm the theoretical concept to produce tiny atomized droplets under a certain vibrational stimulus. However, the theoretical background was shown in a steady state. The formation of atomized droplets should be steadfast and continuous. It is necessary to understand more about the dynamic behavior in ultrasonic atomization for real applications [13]. In this study, a software COMSOL Multiphysics^®^ 5.2 was referenced to understand the formation of atomized droplets by the Computational Fluid Dynamics method. The velocity field distribution was solved by finite-element analysis. In regards to the Faraday instability wave, if the effect of the lateral boundary is ignored, the fluctuations will generate a standing wave at the surface due to vibration at the bottom. The surface vibration that occurred at a high frequency is primarily discussed here. Notably, the essential amplitude for the Faraday instability and the respective wave numbers are observed. It is also notable that the water surface moves with the same flow as the fluid below the surface. In our research, we considered the lateral boundary condition to find the onset amplitude. For the finite consideration, the initial amplitude h_cr_ is dependent on the depth of the liquid. For infinite depth condition, we aimed to determine the initial surface condition when the energy of the capillary surface meets the minimum requirement for atomization. This supports the critical vibrational amplitude (i.e., the characteristic onset of vibrational amplitude) and indicates the constitutive atomized droplets. This work is advantageous for designing a practical device for atomization. 

## 2. Theoretical Background

Faraday instability is characterized by the liquid surface wave in a vessel when the vertical vibrating amplitude reaches the threshold limit. The wave formed on the liquid surface is called the capillary surface wave. Kelvin derived the familiar equation of capillary surface waves when no external excitation was applied to the liquid bottom [14].
λ = [2π*Τ*/ρ*f_s_*^2^)]^1/3^(1)
where *T* is the surface tension, ρ is the liquid density, and *f_s_* is the frequency of the surface waves. It is further stated that the capillary surface wave’s frequency is half of the external frequency. External vibration is applied at the bottom surface of the vessel. In 1883, Rayleigh continued the research and modified (1) as below.
λ = [8π*Τ*/ρ*f*
^2^)]^1/3^(2)
where *f* is the forcing sound frequency, which is empirically two times of *f_s_* found in experimental measurements. Eisenmenger then built a technique for estimating surface viscosity using the theory of rheolinear oscillations [15]. Mathieu’s equation is deformed by transforming the viscosity of liquid under the surface. The minimum capillary wave amplitude is described by the perturbation method as follows: *h_cr_* = 2ν [ρ /(*Τ*ω*_s_*)]^⅓^(3)
where ω_s_ = (*T*k^3^/ρ + gk) ^½^, *k* is the capillary wave number, g is the gravity, *T* is surface tension, and ν is viscosity. Under the low surface viscosity of aqueous solutions and weak vessel boundary conditions, it is further shown that the measured surface capillary wave amplitude is half of the exciting amplitude, with *π*/4 phase delay or shift [16]. The experimental approach is shown by several frequencies in the range from 2 kHz to 800 kHz. Wavelengths reduce as frequency increases, and a good correlation between the size of the atomized droplet and the capillary wavelength at a given frequency is D = 0.34λ, where D is the diameter of the atomized droplet and λ is the wavelength of the capillary waves excited at the certain operating frequency. The theoretical investigation and corresponding experiments on atomization were carried out by Lang for realistic applications [17].

The methanol and glycerol solution was theoretically analyzed with the ultrasonic atomized droplet size and velocity distribution. The ratio between the atomized droplets means diameter and the frequency of the vibration is usually 0.34 ~ 0.36 [18]. Over the last decade, various experiments have been conducted to achieve uniform droplet distribution by varying different parameters, including fluid viscosity, surface tension, water depth, vessel size, vibration frequency, and amplitude. The temperature of the drive section may vary the liquid surface tension and viscosity. A high-magnification camera technique was used to classify the atomization activity at the 32 and 50 kHz working frequencies [18]. Thus, room temperature was considered for the simulation, and the ultrasonic vibrator used for atomization was too low to cause any significant increase in temperature, since the drive used in this study takes advantage of the resonance effect. Prof. Kumar noted a mechanism by which the desired number of waves in the Faraday instability is calculated primarily by Rayleigh–Taylor instability, and the deep-water limit (wavelength < < depth of liquid) is calculated by linear stability [19].

The study of ultrasonic atomization is mostly focused on low-frequency applications. The high rate of increase in Faraday wave amplitude at megahertz drive frequency leads to the onset threshold. In a short amount of time, the monodisperse droplets (>10^7^ droplets/s) start to spray. The measured diameters of the droplets ranging from 2.2 to 4.6 μm at a 2 to 1 MHz drive frequency fall within the optimum particle size range for pulmonary drug delivery [20]. Furthermore, the effects of lateral boundaries are assumed to be negligible when the surfactant is found on the free liquid surface as insoluble, which occurs in the Faraday instability on a surfactant-filled liquid. The insoluble surfactants can potentially lower the value of the critical amplitude relative to its value for an uncontaminated free surface. The phase-angle difference between the free-surface deflection and the surfactant concentration variation showed vibrational amplitude. The minimum critical amplitude rises because the Marangoni flows help to produce a velocity field distribution near the free surface [21]. The formation of Faraday waves in a liquid layer with insoluble surfactant is subjected to a temperature gradient [22,23,24]. The measurements of the frequency corresponding to wave amplitude are still an open question for scientists studying ultrasonic atomization at high frequencies.

To confirm the existence of subharmonic vibration, a frequency spectrum is plotted. Most prominently, the Faraday instability wave was characterized to obtain the minimum energy for atomization. This refers to the design theory of the finite-element analysis for the ultrasonic atomizer. The effect of fluid parameters has also been systematically studied based on Faraday instability. 

Figure 1 is the 2D velocity profile at the corresponding position at 7.5505 × 10^−4^ sec simulation time. The red and blue colors present the displacement in the opposite direction. The lateral velocity profile along the *Y*-axis can be found in the inset of Figure 1, covering the positional velocity of the capillary surface wave.

For the vertical vibration of the vessel with the angular frequency ω and the amplitude *h_e_*, the equation of motion as follows:(4)$$\frac{\partial u}{\partial t}+\left(u.\nabla \right)u=-\frac{1}{\rho }\nabla (P+gz\rho -h_{cr}\omega^{2}z\rho \cos\omega t)+v\nabla^{2}u$$
where u is the velocity field, *P* is the atmospheric pressure, *ρ* is the homogeneous density, and v is the kinematic viscosity. The above equation is the Faraday instability equation. Then, the velocity field into scaler is resolved, and the vector functions are converted into differential equations with variable coefficients by the Helmholtz decomposition [25].
(5)$$u=-\nabla \phi +\nabla \times \vec{\Psi }$$
where ϕ and $\vec{\Psi }$ are the scalar function and vector function of velocity u. $\nabla •\vec{\Psi }=0$. Considering the vibration in a small amplitude and ignoring the nonlinear numerical term, ∇•u=0 as velocity is constant. Thus, $\;\nabla^{2}\phi =0$.

Moreover, based on Bernoulli’s equation, the relation between *P* and *ρ* is as follows:(6)$$\frac{\partial \phi }{\partial t}=\frac{1}{\rho }P+zg-h_{cr}\omega^{2}z\cos\omega t$$

Thus, we define $u_{r}=\nabla \times \vec{\Psi }$, and diffusion velocity u_r_ in the cartesian coordinates as
(7)$$\;u_{r}=u_{rx}\vec{\iota }+u_{ry}\vec{j}+u_{rz}\vec{k}$$
*u_rx_*, *u_ry_*, and *u_rz_* are the velocity components of **u**_*r*_ in x, y, and z-directions, respectively.

Using Equation (6) and **u**_*r*_, the equation of motion is rewritten as follows: (8)$$\frac{\partial u_{r}}{\partial t}=v\nabla^{2}u_{r}$$

This is the standard diffusion equation, in which **u**_*r*_ is the diffusion velocity [26,27]. It is an indicator to analyze the stability of capillary surface waves. In contrast, the system is in a stable situation with a steady fluid surface. In this study, the variation of s was analyzed to understand the initial capillary surface wave. The diameter of the droplets is proportional to the wavelength of the standing capillary waves. The occurrence of the initial instability is another concern; hence, the Floquet index a is considered and assumed as follows: (9)$$\alpha =\bar{\alpha }+m$$
where m is an integral number and $\left|\bar{\alpha }\right|\leq 0.5$ denotes universal values [28]. It is also noted in harmonic vibration mode at α = 0 that the capillary surface wave frequency is found to be equal to the stimulated vibration frequency [29]. The previous study has shown a wave growth rate of s > 0, indicating that the fluid surface could generate waves on the viscous fluid surface [30]. 

Considering a boundary condition up to the free liquid surface, a vertical vibration frequency of 500 kHz was applied at the bottom of the vessel. Liquid depth *h* was equal to 1 mm; the numerical analysis method demonstrated a subharmonic at the capillary surface wave frequency of ω/2, 3ω/2, 5ω/2, etc. The obvious phase difference in the subharmonic mode was noted as 0.26π [31]. The approximated Faraday wave is given as below:(10)$$\eta \left(x,y,t\right)\approx 2{\bar{c}}_{1}e^{i\left(k_{x}x+k_{y}y\right)}\cos(\frac{\omega }{2}t-0.26\pi )$$

Here, k_x_ and k_y_ are components along with the X and Y directions of the wave vector, respectively, and $\bar{c}$ is any real constant. 

Figure 2 shows the integrated results of the Floquet index α = 0 and α = 0.5 of the vertically vibrated vessel at *f* = 500 kHz. The plot at *s* = 0 is a concave upward curve with a minimum at the vertex (3.25 × 10^5^ m*^−^*^1^, 0.349 μm), as illustrated. The vibrational amplitude reached the threshold value of 0.349 μm. Due to a non-zero wave growth rate, the vibration energy was sufficiently stored to produce tiny waves. A Floquet index α was assumed to calculate the corresponding *s* value of the initial capillary surface wave. s > 0 means the surface is unstable, and s < 0 means the surface is stable. The neutral point between stable and unstable is s = 0. First, the standing wave at the surface appeared, then as time elapsed, the amplitude of the capillary surface wave increased above a threshold and broke the surface tension constraint. A sub-harmonic model was used to elucidate the initial vibration behavior so that the fluid liquid at the surface could be atomized.

## 3. Effect of Hydrodynamic on Faraday Wave

The effect of the vibration frequency on h_cr_ and k_cr_ is significant for the selected liquid. The fluid liquid depth was extended to infinity for the simplicity of calculation. Discussing the response of h_cr_ and k_cr_ in the scanned frequencies with hydrodynamic parameters would also be simplified. 

Elevated vibrational amplitude h_cr_ decreased, but the k_cr_ increased as the operating frequency increased, as shown in Figure 3. The integrated results suggest that the corresponding Floquet index at α = 0.5 correlates well with the previous section’s observation for the subharmonic wave mode generation.

The surface wave’s operating frequency is equal to half of the vibration level. The wavenumber k_cr_ dominated at a high operating frequency after 300 kHz. A weak viscosity liquid system was determined to explain the frequency dispersion of the surface wave frequency ω_s_ and k_cr_ [32].
(11)$$\omega_{s}^{2}=k_{cr}^{3}T^{\prime }+gk_{cr}$$

## 4. Finite-Element Analysis

For the finite-element analysis, COMSOL 5.2 software was adopted throughout the modeling to verify the theoretical understanding in the previous section. In this study, COMSOL Multiphysics was referenced to understand the formation of atomized droplets by the Computational Fluid Dynamics method. For fine mesh resolution in COMSOL Multiphysics, 2D simulation computation times became significantly infeasible for numerical analysis. The capillary surface wave was modeled by the laminar flow feature. For simulation of the numerical model, a rotating machinery laminar flow was used to implement flow continuity boundary conditions. Particularly, the flow continuity boundary condition was applied to the identity pair. As there was no rotating machinery in our case, we set ω = 0 to fit our problem. 

For the simulation, the mesh was considered as 1/20th of the dimensions of wavelength. This did not affect the results of the 2D simulation. Since we simulated the Faraday wave, we constructed the grid according to the characteristics of the wave. In the CFD simulation, we adjusted the size according to COMSOL grid length requirements. The comparison simulation and experimental results are also consistent, so 1/20th of the wavelength was sufficient. 

For the user-defined option, we entered a suitable maximum element size in free space. For example, take any case in which 1/5th of the vacuum wavelength or smaller is used. When frequency is selected, we entered the highest frequency intended for use during the simulation.

The maximum mesh element size in free space is 1/5th of the vacuum wavelength for the entered frequency.

For the wavelength option, we entered the smallest vacuum wavelength intended for use during the simulation. The maximum mesh element size in free space was 1/5th of the entered wavelength. When a resolved wave in lossy media was selected, the outer boundaries of lossy media domains were meshed with a maximum mesh element size in free space, which was given by the minimum value of half a skin depth and 1/5th of the vacuum wavelength. In free space, the maximum mesh element size in dielectric media was divided by the square root of the product of the relative permittivity and permeability.

The Navier–Stokes equation is the traditional mathematical tool for finding solutions and understanding the nonlinear effects of liquid water. The frequency of operation is 500 kHz, at which the free surface domain is the boundary state for a subharmonic wave at the fluid surface to measure the amplitude. The forces on the liquid surface obtained as vertical vibration can be determined from the simulation results. A linear stability review was performed on several viscoelastic liquids [33,34]. For simplicity in simulation, the liquid water is suitable for the case study. Our focus was to detect the capillary surface wave onset amplitude under a vertical sinusoidal oscillation in a liquid bottle.

The entire system consists of a water tank, as depicted in Figure 4. For the simulation, operating frequency was set at 500 kHz and the vibrational wavelength was set at 19.3 μm. The tank width L was equal to thirty times the vibrational wavelength as 580 µm, and the water depth *h* was more than six times larger than the vibrational wavelength as 116 μm. Therefore, the density, dynamic viscosity, and surface tension of the background medium (water) were *ρ* = 1000 [kg/m^3^], *v* = 1.02 × 10^−6^ [m^2^/s], and *T* = 7.25 × 10^−2^ [N/m], respectively. The moving wall is the setting of the boundary condition for the bottom of the water tank. The velocity in the Y direction is h_e_xω$x_{\chi }$sin(ωt). The h*_cr_* is a given amplitude, ω *=* 2πf is the angular frequency, and f is the vibrational frequency of the water tank. Thus, the momentum equation is as follows.
(12)$$\rho \frac{\partial u}{\partial t}+\rho \left(u.\nabla \right)u=\nabla .[-PI+\mu \left(\nabla u+\left(\nabla u\right){\;}^{T}\right)-\frac{2}{3}\mu \left(\nabla u\right)I]-\rho \left(g-h_{cr}^{\;}\omega^{2}sin\omega t\right)$$
where homogeneous density ρ is 1000 kg/m^3^, μ is dynamic viscosity, $h_{e}^{\;}$ is given amplitude, P is the equilibrium stress at the free fluid surface, and I is the identity matrix.

The free surface limit is the default, with the goal that the velocity of the liquid fluid surface can be removed from a limited distinction estimate technique in the middle region. The fluid surface at P_a_ atmosphere pressure determined as follows:(13)$$\;\left[-pI+\mu \left\{\nabla u+{\left(\nabla u\right)}^{T}\right\}-\frac{2}{3}\mu \left(\nabla •u\right)I\right]•n=-p_{a}+T\left(\nabla_{T}.n\right)-\nabla_{T}T$$

As $\frac{\partial \rho }{\partial t}+\nabla \cdot \left(\rho u\right)=0$ and ∇·u = 0 for the zero consumption of mass, Equation (12) can be rewritten as follows:(14)$$\rho \frac{\partial u}{\partial t}+\rho \left(u\cdot \nabla \right)u=-P_{a}+T\left(\nabla_{t}\cdot n\right)n-\nabla_{t}T-\rho \left(g-h_{cr}\omega^{2}sin\omega t\right)$$

To estimate an onset vibrational amplitude, a scope of vibrational frequency f of the ultrasonic atomizer is filtered by COMSOL.

## 5. Results

### 5.1. Simulated Result

#### 5.1.1. Vibrational Amplitude h_cr_ = 0.36 μm

The first COMSOL study represented the vibration amplitude at 0.36 μm in Figure 5, and observed a non-uniform droplet distribution. However, there was no noticeable formation of the minimum Faraday wave after the COMSOL simulation spread over 300 periods.

The applied vibrational amplitude was confirmed as below the critical onset point, so it did not atomize the fluid at the surface.

The frequency in the *Y*-axis stayed within a limited range, as depicted in Figure 6. The Fourier coefficient of capillary surface wave frequency stayed in a certain range of kHz. Furthermore, the harmonic frequency observed at 1 MHz led to non-uniform droplet distribution.

#### 5.1.2. Vibrational Amplitude h_e_ = 0.38 μm

The second COMSOL study represented the vibrational amplitude at *h_cr_* = 0.38 μm, as shown in Figure 7.

The illustration of the Faraday wave was enhanced within 300 periods. The sufficient energy was transformed into vibrational amplitude, which led the vibrational amplitude to reach the onset value, and atomized the surface layer on the vibrating surface. 

Likewise, a subharmonic frequency was observed around 250 kHz, since the operating frequency was 500 kHz, as shown in Figure 8. 

As solid evidence, the corresponding wavenumber *k* was 3.25 × 10^5^ m^−1^, which supports the theoretical calculation. In the water tank, when the vibrational amplitude of capillary surface waves surpassed h_cr,_ and a wave growth rate was s > 0, the amplitude increased with time until the liquid was brokwn down into drops, as the surface tension of the liquid cannot sustain the extreme vibration. Similarly, the wavelength of the capillary surface wave was 19.35 μm, as determined from the periodicity in Figure 9. The operating vibrational wavelength of 19.3 μm at 500 kHz corresponds well to the result.

#### 5.1.3. Higher Vibrational Amplitude h_cr_ = 0.40 μm and More

The third study at COMSOL simulated elevated h_e_ = 0.40, 0.45, and 0.50 μm. The Faraday wave’s vibrational amplitude was assumed to be increased as the given value increased. Furthermore, if we define v = 0.02 [m/s] as the speed indicator for the Faraday wave formation, the corresponding periodicity is 90, 42, and 30, respectively. The response to shorter periodicity [35] was corrected to the extent that it exceeded the amplitude of the vibrational threshold at h_cr_ = 0.38 μm, which is shown in previous COMSOL studies.

The third COMSOL study in Figure 10 represented the vibrational amplitude at *h_e_* = 0.40 μm. It was observed that Faraday wave was enhanced within 120 periods, which means vibrational amplitude was above the onset value and atomized the surface layer on a vibrating surface.

Recent progress has been made in the scientific literature on the regulation of particle size atomizers with efficient fluid viscosity [36,37]; nevertheless, the minimum energy of the surface capillary wave (i.e., the characteristic onset of the Faraday instability wave) has yet to be systematically explored. We conducted theoretical calculations and COMSOL simulations to explain the data. Multiphysics research has already been a persuasive method for the creation of novel computer designs. The ultrasonic atomization research suggests the potential future applications of this technology.

## 6. Discussion

In our research, the calculation of the onset by considering the boundary of the cylinder and liquid surface is the main aim. In our earlier theoretical research [29,38], researchers investigated the onset point of amplitude h_cr_ at wave growth rate s > 0. As a result, an MFHN (multi-Fourier horn nozzle) nozzle was used on vibration frequency 500 kHz, and we observed a minimum vibration amplitude of 0.37 μm using the LDV. Here, the fluctuating wavelength of the liquids is smaller compared to the depth of liquids, and depth is considered infinite. In our current study, to measure the vibrational amplitude, the boundary condition was considered as the bottom wall of the water tank and the surface domain of the liquid. We measured minimum critical vibrational amplitude *h_cr_* = 0.38 μm using COMSOL simulation. By considering the boundary of the cylinder, critical vibrational amplitude or onset was found to be 0.38 μm, which is very close to the value of 0.37 μm found in our previous theoretical and experimental studies without considering the cylinder lateral boundary and a numerical analysis with an infinite boundary onset value of h_cr_ = 0.349. This result was in good agreement with the previous onset point. 

## 7. Conclusions

Therefore, this study presents the simulation result of ultrasonic atomization spray with Faraday instability phenomena. The amplitude of the capillary surface wave increased as time elapsed, and then finally broke the constraint of surface tension. This result provides novel perspectives compared to previously reported results. As a consequence, a subharmonic model at 250 kHz frequency showed a higher Fourier coefficient of the capillary surface wave at critical vibrational amplitude 0.38 μm than at 0.36 μm. The tiny waves were initiated from a non-zero positive wave growth rate. Onset was recorded at *h_e_* = 0.38 μm at 500 kHz, and at 250 kHz, subharmonic frequency was in a finite lateral boundary condition. This result is in good agreement with the threshold of the onset point, which was predicted to be h_cr_ = 0.349 μm by numerical analysis and the linear theory of capillary wave atomization mechanism. Atomization increased as the vibrational amplitude of capillary surface waves increased, as mentioned in the study.

The critical vibrational amplitude of capillary surface waves is an important physical indicator, and the result suggests that the surface plane with Faraday instability wave would spray micro-droplets under this onset of the single-frequency ultrasonic atomizer. Notably, the newly designed ultrasonic atomizer showed its characteristics of Faraday instability wave resonance on the fluid surface plane.

## Figures and Tables

**Figure 1 micromachines-12-01146-f001:**
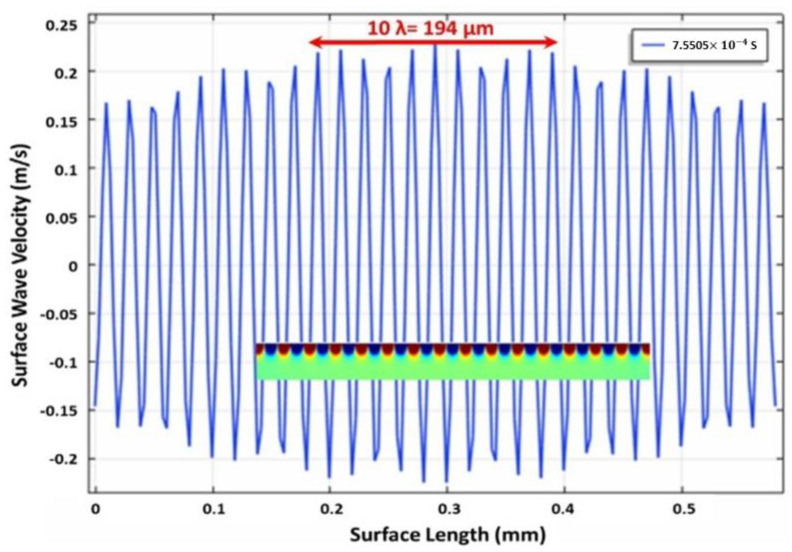
The vibrational velocity is propagated throughout the water surface on the tip of the ultrasonic atomizer in the modeling.

**Figure 2 micromachines-12-01146-f002:**
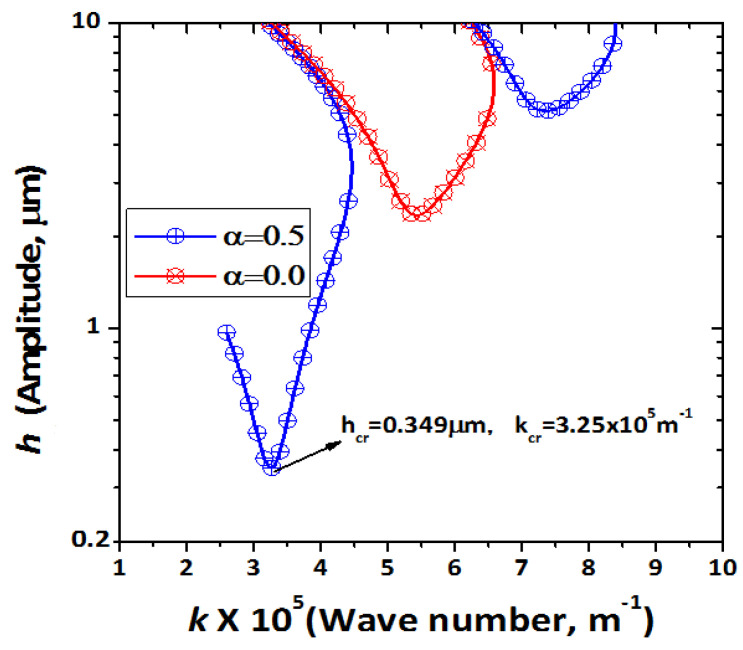
Relationship between the amplitude and the wave number at α = 0 and α = 0.5 for s = 0 at f *=* 500 kHz.

**Figure 3 micromachines-12-01146-f003:**
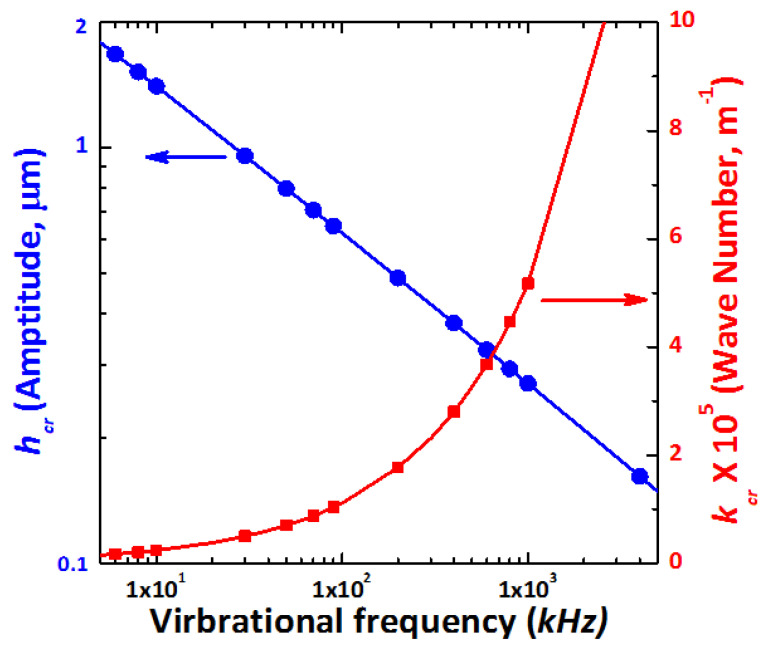
The dependence of the amplitude h_cr_ and the wave number k_cr_ with respect to the operating frequency.

**Figure 4 micromachines-12-01146-f004:**
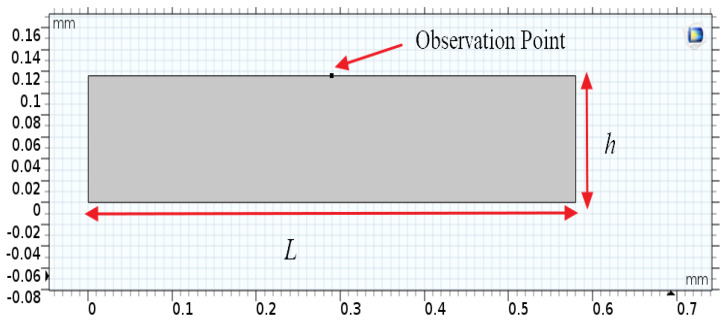
The water tank design is settled in the COMSOL environment. The water tank size: the width L is 580 μm, and the depth h is 116 μm.

**Figure 5 micromachines-12-01146-f005:**
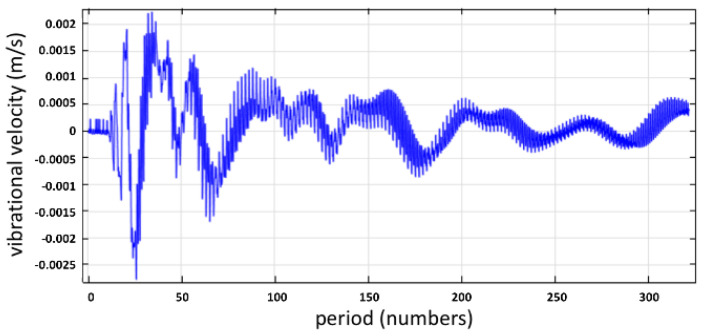
At the position (L/2, h), the vibrational velocity varies with the time period for h_cr_ = 0.36 μm.

**Figure 6 micromachines-12-01146-f006:**
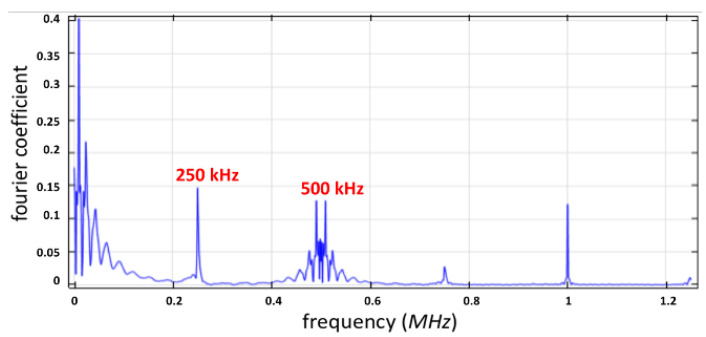
The frequency spectrum at the position (*L*/2, *h*) for *h_cr_* = 0.36 μm.

**Figure 7 micromachines-12-01146-f007:**
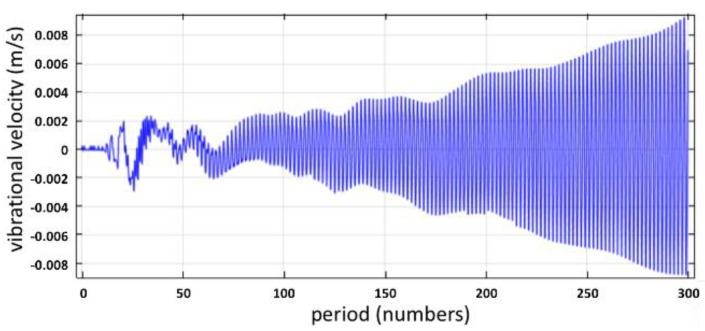
At the position (L/2, h), the vibrational velocity varies with the time period for h_cr_ = 0.38 μm.

**Figure 8 micromachines-12-01146-f008:**
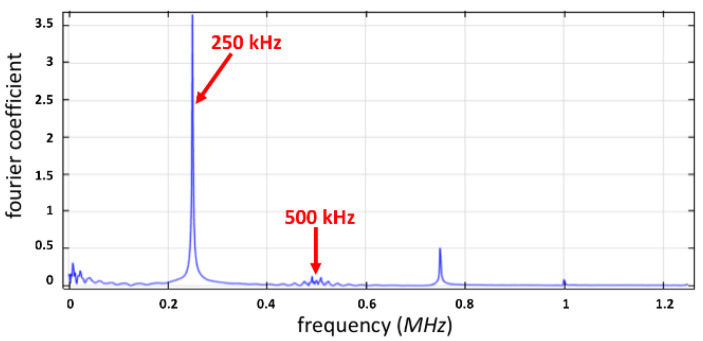
The frequency spectrum at the position (L/2, h) for *h_cr_* = 0.38 μm.

**Figure 9 micromachines-12-01146-f009:**
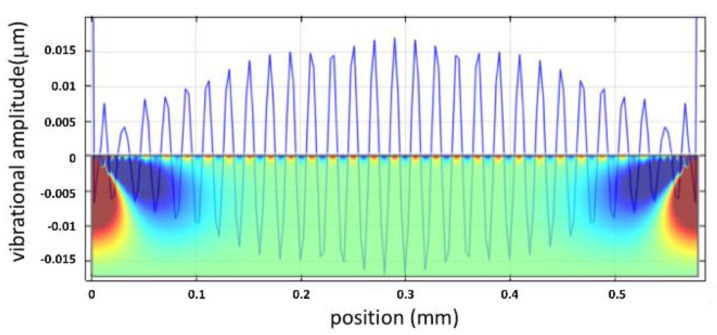
The vibrational amplitude is waved across the water surface for *h*_e_ = 0.38 μm. The inserted figure is the 2D velocity profile in the water tank for 6 × 10^−4^ sec simulation time.

**Figure 10 micromachines-12-01146-f010:**
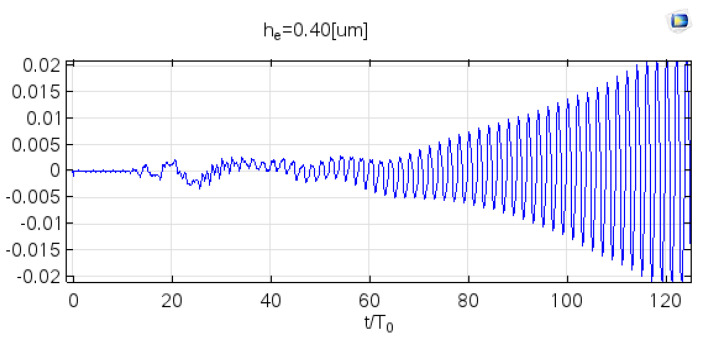
At the position (L/2, h), the vibrational velocity varies with the time period for h_e_ = 0.40 μm.

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
