# Peer review of "Simulation of Onset of the Capillary Surface Wave in the Ultrasonic Atomizer"

_micromachines, 2021, doi:10.3390/mi12101146_

Round 1

Reviewer 1 Report

This paper claims that Faraday wave resonance occurs when a thin liquid layer is oscillated by a piezo actuator with finite amplitude sinusoidal waveforms, and that the resultant Faraday waves break to form monodisperse droplets controllably.

Theory is presented that supports the assertion, but the major support is proposed to come from finite element simulations of the waveforms, using the commercial package Comsol Multiphysics.

The description of the implementation of the model lacks significant detail for anyone to follow.   There are many issues for the numerical simulation of any study that are not addressed:

  1. Self consistency through mesh resolution -- what is the size of the mesh elements vs. the wavelength of the phenomenon?   How do you know that you have sufficient mesh resolution that you are not just analyzing numerical artefacts?
  2. Self-consistency through time stepping?  How do you know that you have sufficient temporal resolution?  Comsol Multiphysics in version 5 has built-in assumptions in time stepping / algorithms that aims to achieve stability and resolution, but these assumptions may lead to wholly irrational results.   The automatic scaling of variables and the implementation of consistent initial conditions may not lead to consistency with the simulation aims.   You could be interpreting dynamics implemented by the Gear's Method / backward Euler time stepping that is known to create non-physical oscillations.
  3. Three studies are considered with fixed parameters, yet the resonance theory is about surface tension, density and viscosity of the liquid presuming an air environment.  There are also geometric parameters.   It is impossible to conclude any understanding of such as high dimensional parameter space from three data points.

Overall, this is a very thin study that is poorly presented -- so violates the primary principle of peer review that peers could duplicate the study -- and the methods are not sufficiently justified.   Any fool can do a calculation.  The trick is knowing why it is correct.   Validations and self consistency studies are just lacking, as well as a sufficient breadth of study so that the claim that resonant waves have the desired properties is justified.   

It is a well known principle of modelling that you "get what you put".   The laminar Navier-Stokes equations never produce turbulence.   Wave breaking is a common turbulence generation mechanism with high amplitude / fast waves.   So fundamentally this study ignores the possibility that the desired effect does not occur due to flow instability generating turbulence.  Harmonic forced damped nonlinear systems are the paradigm for chaotic wave dynamics.  How do you know that your simulation is not producing non-physical results because it is operating in a regime that is dominated by physics not captured by your model?

Reviewer 2 Report

In this article the authors present results of  the simulation of ultrasonic atomizer  with Faraday instability phenomena. The simulation studies were carried out for a specific, individual case basis using standard equations and methods. A number of simplifying assumptions were made, water was taken as the liquid. The result is the determination of the critical amplitude of the vibration excitation. In the opinion of the reviewer, with such assumptions, the results of simulation tests without experimental verification have a low scientific value. The authors should respond to this problem. The more so because this is another article by the first author in this field, and other articles present the physical instrument and experimental results. Therefore, it is incomprehensible to limit ourselves to a standard computer simulation only. The more that the obtained results are not universal, but relate to a specific case.

The formal quality of the article is a separate issue. The drawings come from various sources, they are described with words, not with symbols from equations. Symbols are written heterogeneously. A nomenclature list is necessary. In addition, the article contains errors, for example: "Figure 3 shows the integrated results at Floquet index α = 0 and α = 0 ...", "... the zero consumption of mass, Eq. (5) can be rewritten as .. . ", etc. It is necessary to carefully and formally correct the entire article.

Round 2

Reviewer 1 Report

The authors should note that they have two sections enumerated 6.

The authors have failed to answer several of the criticisms made satisfactorily.

  1. Self consistency / mesh resolution.  The authors state that they cannot achieve a refined mesh for a 2D time dependent study that actually illustrates mesh independence, but specify that 1/20th of the wave length is sufficient.   Until the authors illustrate a mesh independence study, rather than an unsupported claim, their study is unfounded -- equivalent to an experimental study without assessing the experimental variability and replication errors.
  2.  Self-consistency / time resolution.  The authors blandly state that time dependence is taken care of by Comsol Multiphysics 5.2 without addressing the issues raised.  Comsol's default solver is a stiff solver implementing backwards differencing that is appropriate for many scenarios, but can cause oscillations in initial value problems with initially inconsistent conditions.  The built-in solution is to smooth the initial conditions for consistency, which can provide spurious oscillations in transient problems, or start far from the actual solution.  There are a number of ways to overcome this problem, but first you have to assess if you have got it.   The appropriate method is to turn off the stiff and adaptive solver, make sure to "uncheck" the default "consistent initial conditions" dialogue box, and use a fixed time step with an standard solver, such as RK45.  Then compare the major features of your solution to the RK45.   If they agree to a good tolerance, then you can conclude your adaptive / stiff solver is providing sufficient temporal resolution.
  3.  The authors state now "For fine mesh resolution in COMSOL
    Multi-Physics, 2D Simulation computation times become significantly infeasible for numerical simulation. To solve time unsteady flow in Computational Fluid Dynamics,
    time-averaged equations such as the Reynolds-averaged Navier–Stokes equations
    (RANS) are used with high amplitude / fast surface waves."  Then they tell us they are using a laminar flow model appropriate for rotating machinery.   Can they not see that they have just contradicted themselves?  It cannot be turbulent and laminar at the same time, physically, nor are the models the same.   Since it seems they have employed a laminar flow module, which do they need the rotating machinery feature?  What part of their model is rotating?  Where is the machinery?  From my understanding of this module, it is appropriate for, say, an impeller region in a stirred tank overlapping with fluid region that does not have the impeller sweeping through it.  The swept region and unswept region can be treated with different models, with the boundary condition in between maintaining certain continuity conditions.   If they have some use that is non-trivial for this stated purpose of the rotating machinery module, it is certainly not explained.
  4. Validation studies.   The authors have added a comparison to published theory for one figure of merit.  That is a good start.  I support Referee 2's point that the authors have their own experimental studies that they have not used for validation, which is a significant weakness to both sets of work.   Experimental agreement (or regimes of disagreement) place a theoretical and computational model in perspective -- what it gets "right" and what is still unexplained.  Sometimes it can highlight failures in the experimental design.

Reviewer 2 Report

In the opinion of the reviewer, the Authors did not take the submitted comments seriously.
The issue of the experimental verification of the obtained simulation results has not been discussed at all. So what do the authors base the credibility of the obtained results on?
Also, the form of the article was only symbolically improved. Figures are still not standardized. For example, Figure 1. This is not a figure for a scientific article, but only an expression about the phenomenon.
Currently, the article has two chapters number 6! I am asking for a serious correction of the article. Both substantive in terms of experimental verification of numerical results, and formal in terms of quality and error correction. The two errors noted in the previous review were just an example. 

Round 3

Reviewer 1 Report

In the second report, I put four serious issues to be addressed.   The authors have addressed them, but not satisfactorily.

  1.  Self consistency / mesh resolution. 
    Stating that the Comsol manual says it is satisfactory to use 1/20th the wavelength as the mesh is insufficient as a mesh resolution study.  That is a rule of thumb, not a study.  Chose the most important emergent feature predicted by the study, and show that the numerical value is convergent to a reasonable tolerance as the mesh is resolved.   1/20th of the wave length may only achieve 5% accuracy, which could compound errors!
  2. Self-consistency / time resolution.    Showing how Comsol implements BDF and specifying 1/50th of a period as a time step does not show temporal resolution to more than 2%, which again can compound.   It neglects completely the point that the initial conditions can be inconsistent with the boundary conditions -- something usually changes as a discontinuity initially.  BDF does not deal well with discontinuities (it takes "backward" steps, which are discontinuous in the first derivative, for example, with many initial conditions).   My previous report gave a recipe for assessing initial conditions.  Since you are looking for a steady wave solution, a simple approach is to show that after a given time, the power spectrum of your wave is unchanging.   Take a power spectrum at a time t1 and another at a later time t2, typically different by one period, and if the power spectrum is sufficiently similar -- say the power in the maximum mode -- they you can claim a steady wave.   But not to show self consistency with time resolution is indicative of a poor quality study.
  3. Great that the contradiction of claiming laminar and turbulent flows is removed.   Nice to show the Comsol model tree.   But you really should explain in the paper how Comsol implements a free surface boundary condition, and at least tell that it is the air-liquid boundary, initially at height h in Figure 4 that is so treated.   You have also introduced an error by stating \omega = 0 (the rotational velocity in the rotating machinery) and introduced the sine with angular frequency \omega.   If the latter were zero, you do not have a steady wave study.
  4. Really you are not taking the criticism of lack of validation seriously if you are just stating that it is a continuation of previous work.   That hardly is an argument for validation.   

Major revision has been requested by both referees.   You did not provide a major revision, but rather sought excuses for not providing the appropriate required information to meet the standard of a peer reviewed modelling study.    

Reviewer 2 Report

The Authors have correctly answerd most remarks. The article can be published in present form.
